# Effect of Drug Loading in Mesoporous Silica on Amorphous Stability and Performance

**DOI:** 10.3390/pharmaceutics16020163

**Published:** 2024-01-24

**Authors:** Christoffer G. Bavnhøj, Matthias M. Knopp, Korbinian Löbmann

**Affiliations:** 1Pharmaceutical Development, H. Lundbeck A/S, DK-2500 Copenhagen, Denmark; chih@lundbeck.com; 2Bioneer: FARMA, Department of Pharmacy, DK-2100 Copenhagen, Denmark; 3Department of Pharmacy, University of Copenhagen, DK-2100 Copenhagen, Denmark

**Keywords:** mesoporous silica, loading capacity, dissolution, long-term stability, differential scanning calorimetry (DSC), poorly soluble drugs, amorphous stability, surface area, pore volume

## Abstract

The encapsulation of drugs within mesoporous silica (MS) has for several years been a subject of research. Previous studies proposed that drug loadings up to the monomolecular loading capacity (MLC) are the optimal choice for maintaining the drug in an amorphous form, whereas filling the pores above the monolayer and up to the pore filling capacity (PFC) may introduce some physical instabilities. The aim of this study was to assess the effect of drug loading in MS-based amorphous formulations on the stability of the amorphous form of the drug as well as the dissolution. In particular, the following drug loadings were investigated: below MLC, at MLC, between MLC and PFC and at PFC. The drug-loaded MS formulations were analyzed directly after preparation and after 18 months of storage under accelerated conditions (40 °C in both dry and humid conditions). The MLC and PFC for the drug celecoxib (CEL) on the MS ParteckSLC500 (SLC) were determined at 33.5 wt.% and 48.4 wt.%, respectively. This study found that SLC can effectively preserve the amorphous form of the drug for 18 months, provided that the loading is below the PFC (<48.4 wt.%) and no humidity is present. On the other hand, drug loading at the PFC showed recrystallization even when stored under dry conditions. Under humid conditions, however, all samples, regardless of drug loading, showed recrystallization upon storage. In terms of dissolution, all freshly prepared formulations showed supersaturation. For drug loadings below PFC, a degree of supersaturation (DS) around 15 was measured before precipitation was observed. For drug loadings at PFC, the DS was found to be lower and only 6-times compared to the crystalline solubility. Lastly, for those samples that remained amorphous during storage for 18 months, the release profiles were found to be the same as the freshly loaded samples, with similar C_max_, T_max_ and dissolution rate.

## 1. Introduction

The oral formulation and delivery of drugs are preferred due to greater patient compliance and lower production cost [1]. Unfortunately, ever-decreasing drug solubility has become more prevalent over past decades, and many drug candidates face never being commercialized due to the lack of essential solubilization in the gastrointestinal fluids [2,3,4,5]. Therefore, formulation researchers have focused on improving the dissolution rate and solubility of such drugs, and one strategy is loading the drug into/onto mesoporous silica (MS), a strategy that has proven successful in various in vivo studies in rats and dogs but also in humans [6,7,8,9]. MS is generally recognized as safe (GRAS) by the US Food and Drug Administration (FDA) and has been a commonly used excipient in the pharmaceutical industry [10]. Furthermore, viability studies on Caco-2 cells demonstrated no toxic effect in vitro [11].

The drug is loaded into the MS, either via confinement in the mesopores or adsorption onto the surface. Inside the pores, the drug can be regarded as amorphous and, thus, prone to spontaneous recrystallization. However, by loading the drug into an MS grade with pores smaller than the size of the critical nuclei, the initial step of recrystallization (nucleation) may be prevented [12]. Alternatively, adsorption of the drug molecules onto the surface of the MS reduces the molecular mobility, kinetically stabilizing the drug and hindering molecular assembly and clustering [12,13]. For drugs to be successfully loaded on MS, the interaction between the drug molecule and the MS surface should be energetically favorable [14,15,16]. A proposed stabilizing mechanism of the silica is based on surface adsorption by a monomolecular layer, although the drug loading capacity will be substantially lower compared to filling up the pores of the MS. Until 2018, the loading capacity was inconsistently reported due to a lack of both well-defined definitions and tools to distinguish between monomolecular loading and the filling of the pores. Limnell et al. (2011) reported differences in loading efficiency when using two different solvent-based loading techniques: immersion method and a variation of the impregnation method. The immersion method is assumed to provide a monolayer, while the impregnation method enables filling of the pores. Ahern et al. (2013) stated that the loading capacity is related to filling the pores and investigated both solvent-based and non-solvent-based techniques to study the loading efficiency. They found a higher loading efficiency when applying a non-solvent-based technique [17,18]. Using differential scanning calorimetry (DSC), Mellaerts et al. (2007) proposed a method to confirm the monomolecular distribution of itraconazole as the absence of glass transition upon DSC analysis [19]. Later, in 2018, Hempel et al. suggested a procedure to determine the monomolecular loading capacity (MLC) by deliberately filling up the pores with a drug and applying the proportional increase in heat capacity (∆C_p_) from the glass transition when exceeding the monolayer [20]. Hempel et al. further showed that up to the MLC, the loaded drug might be thermodynamically stable. In 2019, the authors additionally showed the relationship between MLC and filling of the pores, i.e., the pore filling capacity (PFC). The otherwise experimentally determined MLC (xMLC) can be estimated with the minimum projected surface of the molecule. This theoretical MLC (tMLC) can be achieved, provided that the estimated PFC is higher than MLC [21]. The MLC estimation was further extended with the maximum projected surface of the molecule by Antonino et al. and, thus, covers an MLC range where the experimentally determined loading capacity can be found. This range illustrated an inherent diversity in the drugs’ molecular packing density on the surface of the MS, which ultimately affects the loading capacity [14]. Antonino et al. also discovered that exceeding the loading past the monolayer can lead to recrystallization. Brás et al. (2011) found, through a spectroscopic analysis, that molecules close to the pore center have a higher mobility than molecules interacting with the surface of the MS [22]. This was also confirmed by Kramarczyk et al., who identified that the stability of celecoxib (CCX) loaded on MS was depend on surface interaction between CCX and MS [15]. Thus, exceeding the MLC can cause drug leakage from the pores, which, subsequently, is prone to recrystallization.

It has also been shown that the degree of drug loading onto the MS influences the drug release from the MS in sink conditions [23]. It is proposed that filling up the pores entirely with a drug will significantly reduce the surface area and will have a proportional impact on the dissolution rate according to Noyes-Whitney [24]. The MLC, therefore, is a safe limit in terms of amorphous drug stability while also maintaining a high surface area to facilitate rapid drug release.

The aim of the present study is to systematically investigate the impact of the degree of drug loading on the long-term amorphous stability and the drug release before and after storage. For this purpose, the model drug CCX was loaded onto the MS ParteckSLC500 (SLC) below MLC, at MLC, between MLC and PFC and at PFC. Subsequently, the samples were analyzed towards their physical stability as well as their dissolution performance on the day of preparation and after 18 months of storage at 40 °C, 0% and 75% relative humidity (RH).

## 2. Materials and Methods

Celecoxib (CCX, M_w_ = 381.4 g/mol) was purchased from Dr. Reddy’s (Hyderabad, India). Sodium chloride, sodium hydroxide, monobasic sodium phosphate and Dimethylsulfoxid (DMSO) were purchased from Merck (Darmstadt, Germany). The mesoporous silica Parteck^®^ SLC 500 (SLC) was received as a gift from Merck (Darmstadt, Germany).

### 2.1. Nitrogen Adsorption

The Brunauer–Emmet–Teller (BET) surface area was determined by nitrogen adsorption on a TriStar 3020 from Micrometrics Instrument Corp. (Norcross, GE, USA). The loaded mesoporous silica, SLC, was dried at 40 °C under a nitrogen purge (1.5 bar) overnight prior to analysis. The sample size was in the range of 15-75 m^2^. The BET specific surface areas were extracted from the linear relationship of 5 points (0.11–0.30 p/p°) via TriStar II software (version 3.02).

### 2.2. Determination of the Monomolecular Loading Capacity and Pore Filling Capacity

The xMLC was determined as described by Hempel et al. [20]. A total of 200 mg of CCX and SLC was physically mixed at the respective drug loadings with mortar and pestle. From the mixtures, 3–5 mg was weighed out into a hermetic aluminum pan and sealed with a lid. Analysis was carried out on a Discovery DSC from TA Instruments Inc. (New Castle, DE, USA). The ∆C_p_ associated with the glass transition was determined by initially annealing the mixtures to 180 °C, well above the melting temperature (T_m_) of CEL, and rapidly cooled to −40 °C, well below the glass transition temperature (T_g_) of CEL. See Table 1 for the T_m_ and T_g_ of CEL. Subsequently, the mixtures were ramped at 20 °C/min to 180 °C. All experiments were conducted in duplicate. Data analysis was conducted using the TRIOS software (version 4.3.0.38) and the ∆C_p_ was plotted against the CCX to SLC ratio in weight % (wt.%). The xMLC was determined as X-intercept (zero ∆C_p_) with a 95% confidence interval using GraphPad Prism (version 7.00).

The tMLC range was calculated based on the minimum and maximum projected surface area of CCX, extracted from the Marvin Sketch software (version 18.10) from Chemaxon (Budapest, Hungary) and the BET surface area of SLC (A_MS_):$$\mathrm{tMLC}=\frac{\frac{A_{\mathrm{MS}}\times M_{w\;\left(\mathrm{CCX}\right)}}{A_{\mathrm{CCX}}\times N_{A}}}{1+\frac{A_{\mathrm{MS}}\times M_{w\;\left(\mathrm{CCX}\right)}}{A_{\mathrm{CCX}}\times N_{A}}}\times 100\%$$
where N_A_ is Avogadro’s number.

The tPFC was determined based on the amorphous density of CCX, reported in [21] and the pore volume of the SLC, provided by the supplier (see Table 1).
$$\mathrm{PFC}=\frac{V_{\mathrm{MS}}\times \rho_{\mathrm{CCX}}}{1+V_{\mathrm{MS}}\times \rho_{\mathrm{CCX}}}\times 100\%\;$$

The xMLC and tPFC were used to define loadings at <MLC (24.3 wt.%), at the MLC (33.5 wt.%), between the MLC and PFC (41.5 wt.%) and at the PFC (48.4 wt.%).

### 2.3. X-ray Powder Diffraction (XRPD)

The solid-state properties of the CCX-loaded SLC were analyzed on a PANalytical, X’Pert Pro diffractometer (Almelo, The Netherlands) with a Κα radiation (1.5406 Å) at a tension of 45 kV and a current of 40 mA. The loaded samples were scanned in the range of 5-40° at a Gonio scan axes with a step size of 0.0167° and scan rate of 2.538°/min. The results were analyzed with the HighScore Plus software (version 3.0).

### 2.4. Non-Sink Dissolution Experiment

To prepare drug-loaded MS material for dissolution, a total of 5 g of CEL and SLC at the respective drug loading was physically mixed with mortar and pestle. The material was spread out on aluminum foil and heated in a Model B 9000 oven from Termaks (Bergen, Norway) at 180 °C for 10 min. The material was mixed again with mortar and pestle and heated in the oven for an additional 10 min at 180 °C. The material was finally powdered with mortar and pestle and the CCX-loaded SLC was analyzed on DSC to confirm the amorphous form of CCX. Neat amorphous CCX was prepared by melting CCX at 180 °C and ground with a mortar and pestle and the amorphous form was confirmed by DSC and XRPD.

The dissolution profiles were examined on a µ-DISS Profiler™ (Pion, Billerica, MA, USA) equipped with 6 fiber optic probes, magnetic stirring at 200 ± 0.3 rpm and 5-/20-mm mirror length. The standard curves and experiments were conducted at ambient temperature (~20 °C). A standard curve was prepared by spiking aliquot amounts of a DMSO stock into the medium; blank FaSSIF buffer (28.65 mM monobasic phosphate, 105.85 mM sodium chloride and 10.50 mM sodium hydroxide, adjusted to pH 6.5). Material was weighed out corresponding to 500 µg of CCX in 20.0 mL blank FaSSIF buffer. The equilibrium solubility of CCX was determined using crystalline CCX as received and by monitoring for consistent plateau. The dissolution curves were examined every 5th second for >30 min. The instrument collects the entire spectrum from 200 to 700 nm with a range in spectrum of 270–320 nm, which was used for CCX. The inherent 2nd derivative function on AuPRO software (version 5.5.3) was applied to correct for particle interference and diffraction with light absorption in both the UV and VIS range.

The maximum concentration (C_max_), time to reach C_max_ (T_max_), area under the dissolution curve (AUC) and dissolution rate were calculated for the individual experiments. The dissolution rate was determined as the slope of the first 60 s of dissolution. A statistical evaluation was conducted using ANOVA test and Student’s *t* test with variance testing. Level of significance was α = 0.05.

### 2.5. Stability and Storage Conditions

The four samples of CCX loaded onto SLC at <MLC (24.3 wt.%), at the MLC (33.5 wt.%), between the MLC and PFC (41.5 wt.%) and at the PFC (48.4 wt.%) were divided into two fractions and stored for 18 months, one at 40 °C/0% RH and the other one at 40 °C/75% RH (open container) obtained with a saturated sodium chloride solution. The loaded samples were analyzed on DSC as described above for presence of T_g_ and T_m_ and subjected to dissolution, as outlined in the non-sink dissolution section above. The diffraction pattern was also confirmed with XRPD.

## 3. Results

### 3.1. Loading Capacity/Drug Loading

The xMLC for CCX loaded onto SLC was found to be 33.5 wt.% (31.7–35.1 wt.%) (Appendix A). This agrees with the calculated tMLC via the minimum projected area of CCX, 33.1 wt.%, suggesting that the CCX molecules occupy a minimal space on the surface of the SLC. The tPFC was determined as 48.4 wt.% via the density of amorphous CCX [21] and the pore volume of SLC (Table 1).

Based on the xMLC and tPFC, four different drug loadings were prepared: below the xMLC (<MLC 24.3 wt.%), at the xMLC (MLC 33.5 wt.%), between the xMLC and PFC (MLC-PFC 41.5 wt.%) and at the PFC (PFC 48.4 wt.%). All four degrees of drug loading were analyzed using DSC to confirm the absence of a glass transition for loadings below and at MLC or the presence of a glass transition but no melting event for samples loaded above MLC but not exceeding the PFC. The amorphousness of the samples was also confirmed by XRPD analysis. The same CCX-loaded samples were analyzed after 18 months’ storage at 40 °C in either a hermetic container (0% RH) or open container at 75% RH. The data are summarized in Table 2 and Appendix A.

To examine the changes in surface area with increased loading of CCX, the BET surface area was determined by the nitrogen adsorption profile. From Figure 1A, it was shown that a linear correlation exists between 23.3 and 41.5 wt.% loading and the BET surface area, i.e., decreasing surface area as more CCX was loaded (r^2^ = 0.99 β_SA_ = −10.1). It was not possible to analyze the surface area of samples loaded to 48.4 wt.% (PFC). At this loading, the surface area would be very small, and the amount of material required would exceed the amounts manufactured. Assuming complete linearity until the pores are filled, the xPFC can be roughly deduced from the interception with the x-axis. From Figure 1A, this can be calculated as 44.0 wt.% and is in close agreement with the tPFC calculated from the amorphous drug density.

The DSC results of CCX loaded onto SLC are summarized in Table 2. As expected, SLC freshly loaded with CCX at 23.3 and 33.5 wt.% (<MLC and MLC, respectively) did not display any thermal events upon DSC analysis when freshly prepared, i.e., no glass transition nor melting event. All loaded samples displayed the expected amorphous halo and lack of Bragg peaks in the XRPD diffractograms, confirming the amorphous form of CCX. XRPD data are not reported but found in Appendix A.

### 3.2. Storage Stability/Physical Stability

As explained above, it has been shown that drugs loaded at or below the MLC do not present a glass transition; however, at higher drug loadings than the MLC, drugs not adsorbed to the surface and found inside the pores display a T_g_ comparable to that of neat amorphous drug. At 41.5 wt.% loading, the amount of CCX should be between the determined MLC and PFC. Overall, the freshly loaded SLC displayed no melting event upon DSC analysis; however, for the loading expected to exceed the MLC, a T_g_ associated with CCX appeared around 60 °C and confirmed that the loaded amount of CCX indeed exceeded the MLC. The samples were divided into two fractions, one subject to 40 °C in a sealed container with drying desiccant (0% RH), and the other at 40 °C/75% RH. Upon 18 months‘ storage, the loaded SLC at 40 °C/0% RH remained amorphous except for the sample loaded at the PFC (48.4 wt.%). Loading the SLC with 48.4 wt.% CCX resulted in the pores being filled. The DSC analysis revealed a T_g_ associated with amorphous CCX as the MLC was exceeded but no melting event attributed to crystalline CCX. Filling the pores of MS has previously been associated with an impairment in the stability as the drug close to the pore entrance is more likely to end outside the pores [14]. This could also be due to inaccurate estimation of the PFC, meaning that the amorphous drug was deposited outside the pores. When stored at high humidity (40 °C/75% RH)and, despite the degree in drug loading, all samples showed recrystallization. The recrystallization was displayed as a melting event in the DSC thermograms and was similar to the melting event of crystalline CCX. The recrystallizations were also confirmed by XRPD, demonstrated as diffraction peaks appearing above the amorphous halo, presumably from the large amount of amorphous silica (Appendix A). The diffraction peaks of the recrystallized CCX matched the diffraction peaks of the starting CCX material. This has previously been reported and attributed to melting of very small crystals of CCX [25]. The samples loaded to 23.3 wt.% displayed a sharp T_m_ upon heating in the DSC at 164.3 °C, but no distinct diffraction peaks were observed in the X-ray diffractograms. This could be due to the lack of sensitivity to trace amounts of crystalline material or could be a result of recrystallization during heating; however, this was not visible in the thermograms. The DSC analysis revealed some evaporation, likely water, which could overlap with the recrystallization upon heating [26]. Loaded samples identified as amorphous upon DSC analysis also proved a halo and no Bragg peaks upon XRPD and confirmed the amorphous form of CCX. Recrystallization at humid conditions (75% RH) is expected as water molecules are known to be a plasticizer in amorphous materials and subsequently accelerate recrystallization [27,28]. Furthermore, the surface of SLC is covered in silanol groups, enabling hydrogen bond formations with the water molecules, which, under humid storage, could compete with the bonds between the SLC surface and CCX [29,30].

### 3.3. Drug Release

The dissolution profiles for the four different drug loadings (23.3, 33.5, 41.5 and 48.4 wt.% CCX), before and after storage, are displayed in Figure 2A–C. The calculated dissolution parameters are summarized in Table 3, and Figure 1B illustrates the AUC_30min_. The freshly loaded SLC improved the dissolution compared to neat amorphous CCX regardless of the loading degree, i.e., higher C_max_, dissolution rate, AUC_30 min_ and shorter T_max_.

The dissolution of neat amorphous CCX displayed significantly lower C_max,_ with an extensively longer T_max_ compared to the SLC-loaded samples. The neat amorphous material was prepared by grinding with a mortar and pestle, which likely had a larger particle size, i.e., smaller surface area, compared to the SLC-loaded samples. The particle size of the latter is expected to be similar to that of pure SLC, i.e., approx. 10 µm. The rapid supersaturation from the SLC-loaded samples can be explained from a contribution of higher available surface area when loaded, enabling it to generate a spring effect. As no precipitation inhibitor was included in the dissolution medium, a parachute effect was not demonstrated. The CCX-loaded SLC enabled a higher degree of supersaturation, which, upon an inflection point, precipitated and lowered the concentrations towards the equilibrium solubility of CCX (1.1 ± 0.1 µg/mL). The samples loaded below PFC were able to achieve a degree of supersaturation (DS) of approx. 15-times prior precipitation. However, when loaded at PFC, the DS was reduced to approx. 6-times before precipitation was detected as a decrease in concentration.

Loading degrees of 23.3, 33.5 and 41.5 wt.% all reached approx. the same C_max_ within the same T_max_, although with a tendency to decrease slightly in C_max_ and increase in T_max_ with increased loading. Despite that, the standard deviation does not allow us to distinguish such differences, and this is somewhat in agreement with classical nucleation theory, stating an early precipitation onset at a higher degree of supersaturation [31]. With increasing drug loading, the dissolution rate decreased, which can be realized from an expected lowering in surface area due to filling up the pores with CCX, also determined from the analysis of BET surface area as a function of CCX loading on SLC (Figure 1A) and also consistent with other studies [25,32]. Regardless of the loading and change in surface area, all three drug loadings (<MLC, MLC, and MLC-PFC) displayed a similar AUC_30 min_ when freshly prepared, indicating that the total release was not impeded.

Upon storage at 0% RH, all three drug loadings (<MLC, MLC and MLC-PFC) remained able to release CCX and supersaturate to the same extent as prior to storage, i.e., similar C_max_, dissolution rate, AUC_30min_ and T_max_. However, higher standard deviation generally applied. As previously mentioned, the stability during storage was impeded by humidity and this was also shown upon dissolution (40 °C 75% RH stored), revealing C_max_, dissolution rate and AUC_30min_ decreasing and T_max_ increasing (Figure 2, Table 3).

When filling up the pores of SLC completely at 48.4 wt.% CCX loading (PFC), the release of freshly loaded CCX was significantly lower compared to the other drug loadings, i.e., lower C_max_, AUC_30min_ and dissolution rate and higher T_max_. Regardless of storing at 0% RH or 75% RH, the PFC sample recrystallized upon storage, which was reflected in a lower C_max_, dissolution rate and AUC_30min_ compared to when freshly prepared (Table 3). This trend was even more pronounced when stored at 40 °C 75% RH, displayed as an even slower dissolution rate than neat amorphous CCX (Figure 2C). A similar observation was also reported by Riikonen et al. 2015, who proposed that a high loading increases the risk of crystalline material being deposited outside the pores [25]. In addition, the CCX located outside the pores can block the release of the drug from deeper within the pores, explaining the generally lower release [33,34].

Overall, the dissolution rate was inversely proportional to the increasing loading degree (r^2^ = 0.96, β_Diss_ = −0.26). Comparing the linear decrease in surface area to that of the dissolution rate, as depicted in Figure 1A, the two tendencies are not parallel in the applied experimental conditions, i.e., the surface area decreases more drastically than the dissolution rate, even when the CCX molecules were loaded monomolecularly. The lack of proportionality ensures that the release mechanism is fundamentally different from simple Noyes Whitney [32,35].

## 4. Conclusions

This study demonstrated that determining the range of drug loading onto the MS is a relevant factor to safely maintain the amorphous form of the drug during storage as well as preserving the drug product performance, even for extended periods, at least under dry conditions. The dissolution profiles in this study displayed a rapid release, able to create a supersaturated solution, which was followed by drug precipitation. Exceeding the MLC only changed the dissolution profile minorly; however, filling up the pores to the PFC dramatically reduced the C_max_ and AUC over the 30 min duration of the dissolution experiment. Upon storage, the model drug, CCX, remained amorphous for at least 18 months when loaded onto MS and stored at elevated temperatures under dry conditions (40 °C 0% RH) if the pores were not filled (i.e., <PFC). This was also reflected in a very similar release profile upon dissolution when compared to freshly loaded SLC, i.e., same rapid dissolution, C_max_ and T_max_. Regardless of the drug loading, all samples recrystallized when stored at elevated temperature and in humid conditions (40 °C 75% RH), likely due to CCX first being replaced by water on the silica surface and subsequently recrystallized on the outside of the silica pores. This was seen regardless of being monomolecularly loaded or when filling up the pores and was also reflected in an impaired release upon dissolution.

## Figures and Tables

**Figure 1 pharmaceutics-16-00163-f001:**
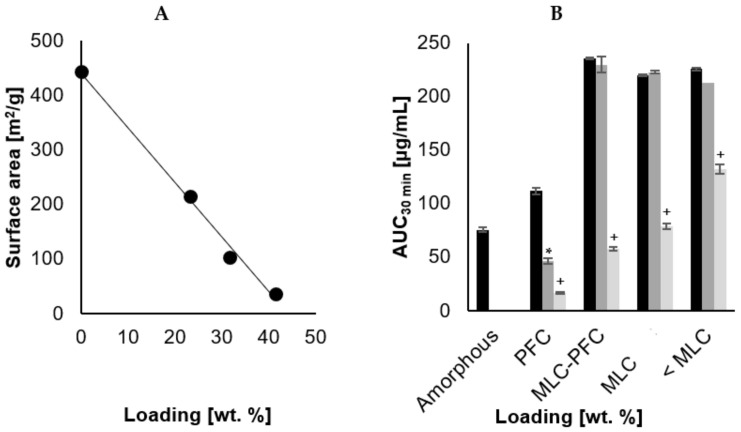
(**A**) illustrates the decrease in BET surface area with increasing drug loading (*n* = 1). The slope of the trendline was β = −10.1 and r^2^ = 0.99. Neat amorphous CCX was tested for 0% drug loading. (**B**) illustrates the AUC after 30 min of dissolution (*n* = 5–6), for the four different loading degrees and neat amorphous CCX [Monomolecular loading capacity (MLC) and pore filling capacity (PFC)]. 

 is freshly loaded CCX, 

 is 18 months stability at 40 °C 0% RH and 

 is 18 months stability at 40 °C and 75% RH. Statistical difference is indicated by * as sign of significant difference between freshly prepared sample and upon storage at 40 °C 0% RH, and + as sign of significant difference between freshly prepared sample and upon storage at 40 °C and 75% RH.

**Figure 2 pharmaceutics-16-00163-f002:**
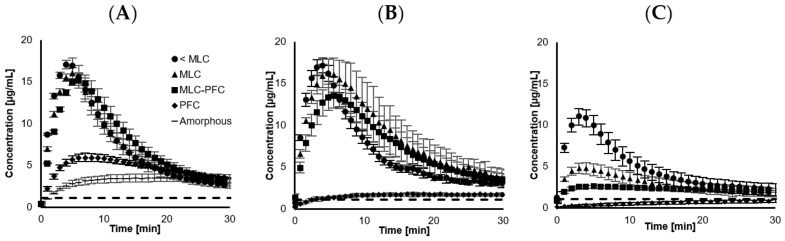
Dissolution curves of CCX loaded onto SLC at different degrees of drug loading (*n* = 5–6) (Monomolecular loading capacity (MLC) and pore filling capacity (PFC)). (**A**) represents freshly prepared samples, (**B**) after 18 months storage at 40 °C and 0% RH and (**C**) after 18 months storage at 40 °C at 75% RH. The dashed line represents the crystalline equilibrium solubility of CCX, 1.1 ± 0.1 µg/mL.

**Table 1 pharmaceutics-16-00163-t001:** Physico-chemical properties of the model compound celecoxib (CCX) and the mesoporous silica Parteck SLC500 (SLC). ∆C_p_ is the heat capacity over the glass transition, ∆H_m_ is the enthalpy of melting, M_w_ the molecular weight and ρ is the density. The standard deviation for the crystalline solubility is denoted with ±.

CCX	T_g_, ∆C_p_	59.0 °C, 0.41 J∙g^−1^∙°C^−1^
	T_m_, ∆H_m_	162.4 °C, 95.9 J∙g^−1^
	Solubility	1.1 ± 0.1 µg∙mL^−1^
	M_w_	381.4 g∙mol^−1^
	ρ_amorphous_	1.35 ^a^ g∙cm^−3^
	Min. proj. area	0.57 ^b^ nm^2^
	Max. proj. area	0.99 ^b^ nm^2^
	tMLC, min/max	22.2/33.1 wt.%
	tPFC	48.4 wt.%
	xMLC	33.5 (31.7–35.1) wt.%
SLC	Surface area ^c^	443.68 m^2^∙g^−1^
	Particle size ^d^	Approx. 10 µm
	Pore volume ^d^	0.73 cm^3^∙g^−1^

^a^ The amorphous density has been reported by [21]. ^b^ The minimum and maximum projection area was provided by Marvin Sketch 18.10 (Chemaxon, Budapest, Hungary). ^c^ Determined by nitrogen absorption, BET. ^d^ The SLC properties; particle size and pore volume were provided by the manufacturer of Parteck^®^ SLC500 (Merck KGaG, Darmstadt, Germany).

**Table 2 pharmaceutics-16-00163-t002:** Results from DSC analysis of the celecoxib loaded onto the mesoporous silica Parteck SLC500 at different drug loadings (monomolecular loading capacity (MLC) and pore filling capacity (PFC)). Samples were analyzed before and after storage for 18 months at 40 °C under (0% RH) and humid (75% RH) conditions. The + or ÷ dictates presence or absence, respectively, of either glass transition temperature, T_g_ or melting point, T_m_.

Loading Degree (wt.%)	Day “0”		18 Months, 40 °C, 0% RH		18 Months, 40 °C, 75% RH	
	T_g_	T_m_	T_g_	T_m_	T_g_	T_m_
PFC	+	÷	+	+	÷	+
MLC-PFC	+	÷	+	÷	÷	+
MLC	÷	÷	÷	÷	÷	+
<MLC	÷	÷	÷	÷	÷	+

**Table 3 pharmaceutics-16-00163-t003:** Dissolution parameters divided by the loading degree and neat amorphous celecoxib (CCX), highest concentration measured (C_max_) and time until this is reached (T_max_). The dissolution of crystalline CCX was not linear during the early dissolution and therefore the dissolution rate was not established.

CCX Loading	C_max_ (µg/mL)	T_max_ (min)	Diss. Rate (µg/mL/min)
PFC (day 0)	6.0 ± 0.5	7.2 ± 0.3	1.9 ± 0.2
40 °C/0% RH 18 mo.	2.3 ± 0.4	32.7 ± 10.2	0.5 ± 0.1
40 °C/75% RH, 18 mo.	1.0 ± 0.2	58.7 ± 2.4	Not detectable
MLC-PFC (day 0)	15.4 ± 0.7	5.9 ± 0.5	5.0 ± 0.1
40 °C/0% RH 18 mo.	13.8 ± 1.6	5.8 ± 0.6	4.7 ± 0.6
40 °C/75% RH, 18 mo.	2.6 ± 0.2	4.8 ± 0.5	1.3 ± 0.0
MLC (day 0)	16.1 ± 0.7	4.9 ± 0.3	6.8 ± 0.4
40 °C/0%RH 18 mo.	17.3 ± 1.9	5.3 ± 1.3	7.1 ± 1.1
40 °C/75% RH, 18 mo.	4.9 ± 0.7	3.7 ± 0.3	2.6 ± 0.4
<MLC (day 0)	17.3 ± 0.6	4.4 ± 0.5	8.7 ± 0.2
40 °C/0% RH 18 mo.	17.6 ± 0.6	4.1 ± 0.6	9.1 ± 1.1
40 °C/75% RH, 18 mo.	11.1 ± 0.9	3.4 ± 0.3	6.0 ± 0.8
Amorphous CCX (day 0)	3.6 ± 0.5	20.2 ± 2.0	0.8 ± 0.0
Crystalline CCX	1.1 ± 0.1	Not determined	Not determined

## Data Availability

Data are contained within the article and Appendix A.

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
