# Peer review of "Effect of Drug Loading in Mesoporous Silica on Amorphous Stability and Performance"

_pharmaceutics, 2024, doi:10.3390/pharmaceutics16020163_

Round 1

Reviewer 1 Report

Comments and Suggestions for Authors

Dear respected editor

In the review entitled “Effect of drug loading in mesoporous silica on amorphous stability and performance" reported the effect of drug loading on the stability of the mesoporous silica with respect to dissolution study and storage

The article is quite interesting and can be accepted for publications after minor corrections The important points to the author are listed below:

1-     First of all the manuscript should be checked by an English native speaker to remove the syntax and typos.

2- The abstract should be modified to give more digital results rather than elastic sentences. 3- The numbering of the titles and subtitles should be adjusted

4- The authors should state the method of drug loading in the beginning of the methodology

5- The authors should identify the word “sample” all over the manuscript, the methodology is so misleading

6- The standard deviation bars should be added to the figures

7- The authors should use symbols as *, + on all figures to show the statistical differences

Comments on the Quality of English Language

Dear respected editor

In the review entitled “Effect of drug loading in mesoporous silica on amorphous stability and performance" reported the effect of drug loading on the stability of the mesoporous silica with respect to dissolution study and storage

The article is quite interesting and can be accepted for publications after minor corrections The important points to the author are listed below:

1-     First of all the manuscript should be checked by an English native speaker to remove the syntax and typos.

2- The abstract should be modified to give more digital results rather than elastic sentences. 3- The numbering of the titles and subtitles should be adjusted

4- The authors should state the method of drug loading in the beginning of the methodology

5- The authors should identify the word “sample” all over the manuscript, the methodology is so misleading

6- The standard deviation bars should be added to the figures

7- The authors should use symbols as *, + on all figures to show the statistical differences

Author Response

1- First of all the manuscript should be checked by an English native speaker to remove the syntax and typos.

Authors reply: We have checked the manuscript and updated the English language.

2- The abstract should be modified to give more digital results rather than elastic sentences. 

Authors reply: We have update the abstract according to the reviewers suggestion.

3- The numbering of the titles and subtitles should be adjusted

Authors reply: The numbering has been updated.

4- The authors should state the method of drug loading in the beginning of the methodology

Authors reply: The methodology section has been moved to the beginning of the section.

5- The authors should identify the word “sample” all over the manuscript, the methodology is so misleading

Authors reply: We changed the wording for several cases and changed some for loaded samples.

6- The standard deviation bars should be added to the figures. The authors should use symbols as *, + on all figures to show the statistical differences.

Authors reply: standard deviations are present on the figures but are narrow. Statistical indications: *, + has been added to figure 1B.

Reviewer 2 Report

Comments and Suggestions for Authors

The authors present a study of the drug loading dependence of celecoxib (CCX) in mesoporous silica (MS) and its impacts on stability and performance.  This work builds on literature in the study of celecoxib in various types of mesoporous silicas with a systematic study of the loading dependence of CCX between the monolayer adsoption limit and the pore filling limit.  This work is unique from prior literature in two respects, 1) few studies have examined the properties of MS in a systematic way between well-defined loading limits, and 2) this work demonstrates the long-term stability properties of CCX-loaded MSs at the various loadings.  This information should be valuable to researchers progressing MSs as a platform.

Comments on the manuscript:

Two papers are cited on the properties of CCX in MS, yet there is an extensive literature on this system, including one recent article by Kramarczyk et al. on the temperature dependent molecular dynamics of CCX in MS that may be relevant to the stability results of this work.  I'd encourage the authors to include a more thorough review of the CCX/MS literature in the introduction.

Please include more detail in the materials and methods section to describe sample preparation so that other researchers may reproduce these formulations. For example, though a reference is included in the loading dependent experiments to determine the xMLC, no details of how the physical mixtures were prepared and handled.  For example, how was the physical mixture mixed for DSC experiments and at what scale?

In preparation of material for dissolution experiments, was any assessment of particle size made after powdering the samples following preparation in the oven?  Was the material sieved prior to analysis to eliminate agglomerates? 

It is also mentioned in the discussion that slow dissolution of amorphous CCX may be attributable to particle size impacts.  Considering that no normalization of particle size ranges was performed between the CCX-MS samples, is there any chance that those results may also have particle size impacts that may confound interpretation of the results?

Neat amorphous CCX as well as CCX-loaded MS was prepared by 'melting' at 130C, but the melting point of CCX given in Table 1 is 162C.   How can this be? 

Description of the stability study in the materials and methods section should note that both open and closed conditions were used.  Also, please describe the "four samples" in this section as well.

In the stability discussion, it would be helpful to understand T/Tg for the stability conditions selected as this is indicative of the thermodynamic and kinetic conditions governing crystallization.  In the 40C 0%RH condition the T/Tg ratio is ~0.94, and reasonable physical stability is observed at loadings below the PFC limit.  This is consistent with the temperature being below the Tg, and the formulation remaining in the glassy state. However, at the 40C/75%RH condition, a marked degradation in amorphous state and performance is noted.  This condition may have a T/Tg ratio exceeding "1", meaning that CCX could have been above its Tg and crystallization may have commenced due to high molecular mobility conditions, in addition to a greater thermodynamic driving force for crystallization.  One useful reference for Tg vs. RH for CCX is " Shete, Ganesh & Kuncham, Swathi & Puri, Vibha & Gangwal, Rahul & Sangamwar, Abhay & Bansal, Arvind. (2014). Effect of Different “States” of Sorbed Water on Amorphous Celecoxib. Journal of Pharmaceutical Sciences. 103. 10."

Comments on the Quality of English Language

Very minor corrections needed.  Text editor check should be able to identify issues.

Author Response

1- Two papers are cited on the properties of CCX in MS, yet there is an extensive literature on this system, including one recent article by Kramarczyk et al. on the temperature dependent molecular dynamics of CCX in MS that may be relevant to the stability results of this work.  I'd encourage the authors to include a more thorough review of the CCX/MS literature in the introduction.

Authors reply: We have update the cited literature and added more recent references to the manuscript.

2- Please include more detail in the materials and methods section to describe sample preparation so that other researchers may reproduce these formulations. For example, though a reference is included in the loading dependent experiments to determine the xMLC, no details of how the physical mixtures were prepared and handled.  For example, how was the physical mixture mixed for DSC experiments and at what scale?

Authors reply: We have added more details to the method section.

3- In preparation of material for dissolution experiments, was any assessment of particle size made after powdering the samples following preparation in the oven?  Was the material sieved prior to analysis to eliminate agglomerates? 

Authors reply: No particle size measurements were conducted. No agglomerates were observed neither in the freshly prepares samples nor after storage. The loaded silica had same “flowability” as non-loaded silica.

4- It is also mentioned in the discussion that slow dissolution of amorphous CCX may be attributable to particle size impacts.  Considering that no normalization of particle size ranges was performed between the CCX-MS samples, is there any chance that those results may also have particle size impacts that may confound interpretation of the results?

Authors reply: The dissolution of amorphous CCX was only done for comparison. The main focus was to study the mesoporous based drug loaded amorphous systems. We have added one sentence to the discussion to highlight that the particle size of the loaded material was assumed to be similar to those of pure SLC.

5- Neat amorphous CCX as well as CCX-loaded MS was prepared by 'melting' at 130C, but the melting point of CCX given in Table 1 is 162C.   How can this be? 

Authors reply: We would like to thank the reviewer for pointed this out. This was a mistake in the original submission and has been changed now. The preparation is done at 180°C.

6- Description of the stability study in the materials and methods section should note that both open and closed conditions were used.  Also, please describe the "four samples" in this section as well.

Authors reply: this has been done.

7- In the stability discussion, it would be helpful to understand T/Tg for the stability conditions selected as this is indicative of the thermodynamic and kinetic conditions governing crystallization.  In the 40C 0%RH condition the T/Tg ratio is ~0.94, and reasonable physical stability is observed at loadings below the PFC limit.  This is consistent with the temperature being below the Tg, and the formulation remaining in the glassy state. However, at the 40C/75%RH condition, a marked degradation in amorphous state and performance is noted.  This condition may have a T/Tg ratio exceeding "1", meaning that CCX could have been above its Tg and crystallization may have commenced due to high molecular mobility conditions, in addition to a greater thermodynamic driving force for crystallization.  One useful reference for Tg vs. RH for CCX is " Shete, Ganesh & Kuncham, Swathi & Puri, Vibha & Gangwal, Rahul & Sangamwar, Abhay & Bansal, Arvind. (2014). Effect of Different “States” of Sorbed Water on Amorphous Celecoxib. Journal of Pharmaceutical Sciences. 103. 10.

Authors reply: We have not determined the Tg of the confined CCX under humid conditions. It is true the Tg is plastisied under humid conditions which facilitates the recrystallization. However, for the confined CCX (<PFC) the material shall be prevented from crystallization by the confinement in the mesopores, which in theory are smaller than a crystal nuclei. Hence, the drug cannot crystallize inside the pores, however, it can flow out of the pores and then crystallize. This process was observed in the investigated systems between MLC and PFC. For the systems below MFC, we also observed recrystallization, which can be explained by water replacing the drug from its binding site on the silica surface. Once replaced the drug can crystallize on the outside of the pores as observed in this study. The latter effect is also responsible for the drug release from these systems during dissolution. We have not changed the manuscript, since this explanation was already mentioned in the original submission of the manuscript.

Reviewer 3 Report

Comments and Suggestions for Authors

The abstract should be containing some numerical values of the results.

Please check the manuscript for minor grammar mistakes.

Please, add the aim of the study in s clear sentences at the end of introduction section.

Please, check the use of acronyms throughout the manuscript.

Please, add paragraph at the  of materials and methods about statistically analysis including software used in analysis of data, data expression, p values.

Each figure and table should be self presentable in the term of acronyms, samples number, data expression, ect.

This work lacks the biological experiment.

The reference list must updated, the most recent reference is 2020. Please, address this issue and add 2021, 2022, 2023, 2024 in the reference list.

Comments on the Quality of English Language

Minor grammatical errors.

Author Response

1- The abstract should be containing some numerical values of the results.

Authors reply: The abstract has been changed with some numerical values. The result section has been changed accordingly.

2- Please check the manuscript for minor grammar mistakes.

Authors reply: This has been done.

3- Please, add the aim of the study in s clear sentences at the end of introduction section.

Authors reply: The wording has been changed at the end of the introduction to highlight the aim of the study.

  1. Please, check the use of acronyms throughout the manuscript.

Authors reply: This has been done.

5- Please, add paragraph at the of materials and methods about statistically analysis including software used in analysis of data, data expression, p values.

Authors reply: standard deviations are present on the figures but are narrow. Statistical indications: *, + has been added to figure 1B.

6- Each figure and table should be self presentable in the term of acronyms, samples number, data expression, ect.

Authors reply: each figure and table has been reviewed and changed to better explain.

7- This work lacks the biological experiment.

Authors reply: We fully agree with the reviewer’s comment that a biological evaluation in form of a PK study would be highly interesting and important whether the in vitro findings in fact do translate into similar observations in vivo. However, before conducting any PK studies, different formulations typically are analyzed in vitro to assess their performance. This was precisely the main purpose of this study, i.e. to compare the in vitro dissolution profiles of mesoporous based amorphous formulations when freshly prepared and stored under accelerated conditions. However, the authors believe this was out of the scope of this study, and hence, was not investigated.

8- The reference list must updated, the most recent reference is 2020. Please, address this issue and add 2021, 2022, 2023, 2024 in the reference list

Authors reply: We have looked at more recent literature and added throughout.

Reviewer 4 Report

Comments and Suggestions for Authors

This paper reported the effect of drug loading in mesoporous silica on amorphous stability and performance. Revisions are suggested based on the following questions.

1. Basic animal and cellular evaluation results (at least basic cellular evaluation results) should be presented.

2. It is necessary to provide basic physical and chemical property evaluation results before and after drug loading.

Author Response

1- Basic animal and cellular evaluation results (at least basic cellular evaluation results) should be presented.

Authors reply: We have updated the introduction with published information concerning the safety of mesoporous silica.

2- It is necessary to provide basic physical and chemical property evaluation results before and after drug loading.

Authors reply: In the context of the aims of the study, we have done the relevant characterization of the investigated materials, including XRPD, DSC and dissolution. The relevant evaluation for MS was also provided in table 1.

Round 2

Reviewer 1 Report

Comments and Suggestions for Authors

the manuscript has been sufficiently improved to be published

Reviewer 4 Report

Comments and Suggestions for Authors

The animal and cellular drug releases profile should be presented.